# The Artificial Mixed Fused Quartz Particles and Silicon Particles-Assisted High-Performance Multicrystalline Silicon

**Yanhuan Cai [1], Changcheng Mi [1] and Xinming Huang [1,2,*]**

[1]  College of Materials Science and Engineering, Nanjing Tech University, Nanjing 210009, Jiangsu, China; caiyanhuan1368@163.com (Y.C.); michangcheng822@126.com (C.M.)
[2]  JA Solar Holdings Co., Ltd., Feng Tai District, Beijing 100160, China
*  Correspondence: huangxm@jasolar.com; Tel.: +86-1338-295-6166

**Abstract:** Mixed seeds of fused quartz particles and silicon particles were laid at the bottom of the crucible to assist the growth of multicrystalline silicon crystals. The full melting process was used, and then we found that the grown crystals had higher quality. The effect of mixed seeds on the growth of multicrystalline silicon was studied. The results showed that fine and uniform initial grains could be obtained by mixed seeds assisting the growth of crystals. Increasing the number of grain boundaries can better release thermal stress and inhibit the proliferation and diffusion of dislocations. The defect density of multicrystalline silicon decreased and the minority carrier lifetime increased, thus improving the conversion efficiency of multicrystalline silicon cells.

**Keywords:** fused quartz particles; silicon particles; mixed seeds; conversion efficiency

## 1. Introduction

At present, crystalline silicon occupies more than 90% of the photovoltaic market [1]. Among crystalline silicon materials, monocrystalline silicon (CZ-Si) and multicrystalline silicon (mc-Si) are widely used. The latter is more popular because of its low cost, high output and large-scale advantages. However, compared with CZ-Si, mc-Si crystals contain more defects, such as dislocations (dislocation clusters), grain boundaries (GBs) and precipitations [2], so the corresponding solar cell efficiency is lower.

The traditional multicrystalline silicon is generated with seedless growth. After melting, the silicon raw materials nucleate randomly at the bottom of the crucible, and the initial grains are disorderly. Later, seed-assisted multicrystalline silicon, also known as high performance multicrystalline silicon (HPMC-Si) technology, was developed [3,4]. At present, HPMC-Si has gradually replaced conventional multicrystalline silicon as the most important photovoltaic material by virtue of its excellent product performance. The so-called HPMC-Si can reduce dislocation and improve crystal quality by using seed-assistance to control nucleation and grain growth [4–6]. Full melting and semi-melting are two main processes in HPMC-Si. Fused quartz particles are used as seeds in the full melting process and silicon particles are used as seeds in the semi-melting process. Quartz seeded growth of mc-Si ingots has recently attracted much attention because of the very high yield and high crystal quality [6,7]. Because fused quartz particles are heterogeneous and have no crystal structure and poor infiltration of silicon crystal, the nucleation rate of crystal silicon on its surface is relatively low [8]. The effect of crystallization and the quality of cast silicon ingots are slightly inferior to that of the semi-melting process, while the yield of the semi-melting process is not high due to partial melting of seeds. The full melting process not only has the advantage of yield, but also it does not need seed protection (semi-melting needs to

control the thickness of seeds, high requirements for thermal field, etc.). The operation process is more convenient, and it has become the mainstream process in production. Yin Changhao et al. [9] used SiC materials with better wettability to cast ingots, but the actual effect of the quality of silicon ingots did not be improved as imagined, it became worse. This is thought to be because the introduction of SiC will increase the carbon content in the ingot, reduce the minority carrier lifetime and widen the red region (the low minority carrier lifetime region) at the bottom of the ingot.

Inspired by Wong et al. [10], in this paper, we artificially designed the mixed seeds of silicon particles and fused quartz particles as the nucleation layer. The full melting process was used and then we found that the crystals grown had higher quality than ingots using fused quartz particles. We studied the effects of the special nucleation layer on the quality of multicrystalline silicon ingots and tried to understand the mechanism of defect reduction after using this design. Due to the special nucleation layer at the bottom of the crucible, the quality of crystal was improved. The results showed that, with the help of mixed seeds, uniform and fine initial grains could be easily obtained at the initial stage of crystallization. The seed-assisted multicrystalline silicon based on this method had fewer dislocations and higher conversion efficiency.

## 2. Experimental

### 2.1. Preparation of Seeds and Coatings

The experimental ingots seeded by ordinary fused quartz particles and mixed seeds (fused quartz particles: silicon particles = 1:1) were respectively cast in two directional solidification furnaces of Jingyuntong (JZ460/660). G6 high purity and high efficiency quartz crucible with a size of 1040 mm x 1040 mm x 540 mm was used in the experiment. The purity of the two kinds of particles was 99.9999% and the particle size was 50–70 meshes (200–300 μm). The proper size of the particles was used to prevent the gap between the particles being too large to produce tape casting phenomena [11] or too small to increase the risk of sticking pot.

Seeds were evenly fixed to the bottom of the crucible using an automatic particle planter, as shown in Figure 1. After the seeds were fixed, $Si_3N_4$ coatings needed to be covered. We sprayed two layers of $Si_3N_4$ coatings on the seed layer and the side wall of the crucible using automatic spraying apparatus to make the coatings more uniform. The purpose of $Si_3N_4$ coating is to slow down the erosion rate of molten silicon on the crucible and prevent adhesion between ingot and crucible.

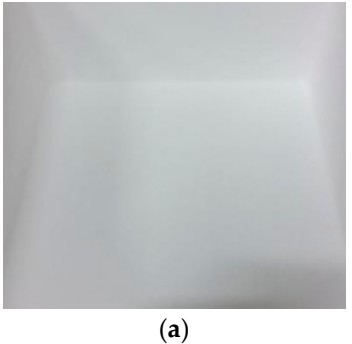 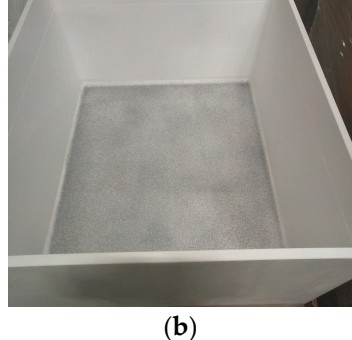

(**a**)                     (**b**)

**Figure 1.** Morphology of the crucible with (**a**) fused quartz seeds and (**b**) the mixed seeds.

### 2.2. Casting Process and Analysis

As a comparison, the technological parameters of the two ingots were consistent. In order to avoid dendrite nucleation at the bottom, the undercooling at the bottom was controlled below 10 k, so the cooling rate at the bottom of crucible was adjusted to about 3 k/min, and the growth rate of the silicon ingot was 12 mm/h.

After casting, the ingot was cut into 6 × 6, totalling 36 silicon blocks, as shown in Figure 2. The initial grain morphology at the bottom of the squares was observed using an infrared detector and a digital camera. The minority carrier lifetime of six blocks B13–B18 was scanned using the minority carrier lifetime instrument (Semilab, WT-2000). Then, the squares were cut into wafers, and the grain morphology and dislocation distribution were characterized by photoluminescence (PL) equipment (BT imaging, LIS-R2). Fabrication and performance testing of the solar cells were performed by Yangzhou JA Solar Technology Co., Ltd.

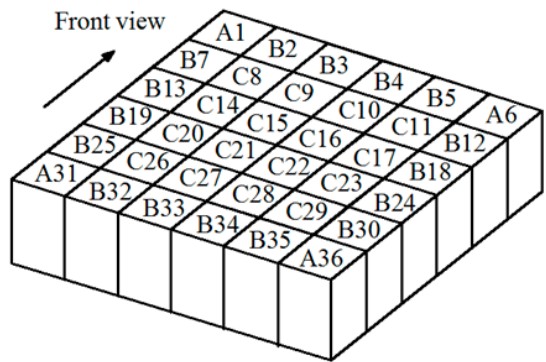

**Figure 2.** Squaring picture of an ingot.

## 3. Results and Discussion

### 3.1. Effect of the Mixed Seeds on the Nucleation Process

The nucleation process is very important. The quality of nucleation directly affects the growth of initial grains. Therefore, it can be said that nucleation is the key factor to improve the quality of silicon ingots. In Ding et al.'s paper [12], two nucleation modes were mentioned. One was the contact between fused silicon and fused quartz particles, which formed a large number of nucleation cores. This nucleation mode was called "contact nucleation". The other was the gap area between fused quartz particles, which was called "gap nucleation". In this experiment, there were also two kinds of nucleation modes. When the silicon particles in the nucleation layer melt at high temperature, there would be some interspace, which allowed liquid silicon to flow into the interspace and solidify and form bead-like homogeneous seeds to co-crystallize with fused quartz particles. Figure 3 is the bottom morphology after demoulding and Figure 4 is the nucleation process.

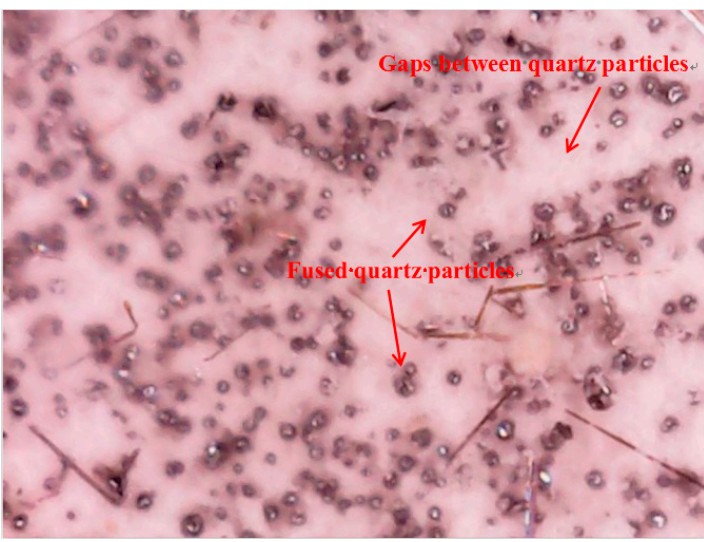

**Figure 3.** Bottom Morphology after Demoulding.

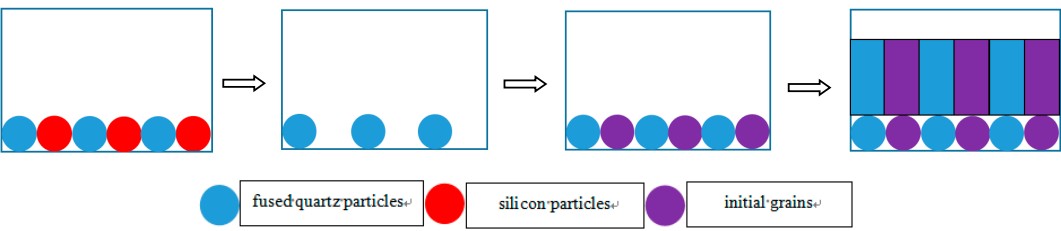

**Figure 4.** Nucleation process.

### 3.2. Effect of the Mixed Seeds on the Grain Size

Figure 5 shows the infrared scanning images of B13–B18 silicon blocks in two ingots seeded by different seeds. In comparison, the grains on the left side (Figure 5a) are relatively loose and nonuniform, while the grains on the right side (Figure 5b) are more vertical and uniform. In order to further compare the initial grain size, two silicon ingots were sampled at the bottom of 2 mm, and the grain morphology in the wafer was photographed by digital camera, as shown in Figure 6. From Figure 4 it can be seen that the initial grain size at the bottom of fused quartz seeded ingot in Figure 6a was relatively large, while that at the bottom of mixed seeds seeded ingot in Figure 6b was obviously small. We used Image-Pro analysis software to measure the size of initial grains. According to the statistics on grain size distribution, the average grain size of fused quartz wafers is about 3.36 mm, while that of mixed seeds wafers is about 2.79 mm, the grain size is reduced about 17.0%, and the grain size is more uniform. Generally speaking, fine and uniform grains in the initial stage are beneficial to suppress the growth of dislocations in the later stage, because grain boundaries have a significant blocking effect on the propagation of dislocations [13], and can better release thermal stress. However, the existence of these grains of different sizes in fused quartz particle-seeded ingots seriously reduces the uniformity of grain size, which is not conducive to the suppression of dislocation in crystals, thus affecting the overall quality of silicon ingots.

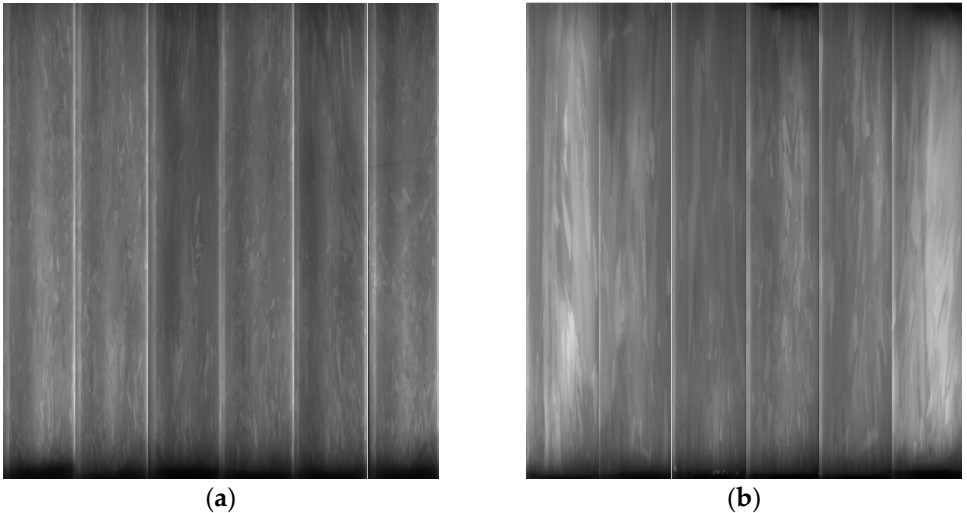

(**a**)  (**b**)

**Figure 5.** Infrared scanning images of longitudinal grain distribution of the bricks B13–B18. (**a**) Fused quartz particles seeded ingot; (**b**) mixed seeds seeded ingot.

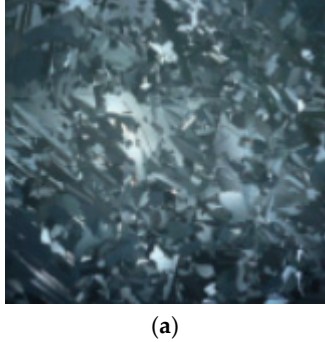

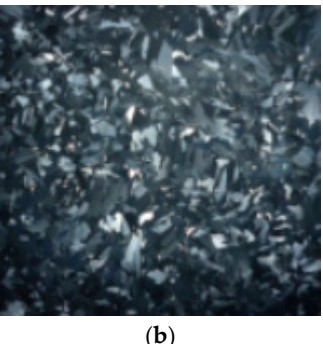

(**a**)    (**b**)

**Figure 6.** Morphology of the initial grains on the bottom of the ingots. (**a**) fused quartz particles seeded ingot and (**b**) mixed seeds seeded ingot.

*3.3. Effect of the Mixed Seeds on the Minority Carrier Lifetime*

Minority carrier lifetime directly affects the conversion efficiency of solar cells. Figure 7 shows minority carrier lifetime maps of B13–B18 squares in ingots seeded by different seeds. The red part in Figure 7 is the low minority carrier lifetime zone, commonly known as the red zone (minority carrier lifetime is less than 2 μs), which is mainly caused by impurity permeation in the crucible, coating and impurities (mainly metallic iron impurities) contained in seeds. Because of too many impurities in the red zone, it cannot be used and needs to be removed and purified in subsequent processes.

The average minority carrier lifetime of fused quartz squares in Figure 7a is 5.78 μs, while that of mixed seeds squares in Figure 7b is 6.20 μs, with a difference of 0.42 μs. In the figure it can be seen that the minority carrier lifetime of the mixed seeds seeded ingot is higher and more uniform, while a large "waterfall" (in red circle) low minority carrier lifetime zone appeared in the upper and middle part of the fused quartz seeded ingot. This may be due to the excessive growth of dislocations in the middle and later stages of the crystal, which resulted in the high distribution of dislocation density, thus forming more minority carrier recombination centers and reducing the minority carrier lifetime.

The color of the red zone at the bottom of the silicon ingot cast with mixed seeds was darker, which indicated that the minority carrier lifetime was lower and the corresponding impurities were more numerous. This phenomenon is in line with the impurity reflux theory proposed by Gaobing et al. [14]. They thought that iron impurity was the main factor leading to the red zone. The impurity iron in the melt flowed back to the seeds and increased the impurity concentration at the bottom, which resulted in an iron concentration at the solid-liquid interface higher than the original concentration, and then diffused to both sides to expand the width of the red zone, which indirectly reduced the impurity content in the silicon melt. This also explained the fact that the minority carrier lifetime of the yellow line in the middle of the red zone was higher than that in both sides (as shown in Figure 7). For the single phase, fused silica sand was a continuous phase with fewer voids, causing fewer reflux of fused silica and fewer impurities. For the mixed phase, the melting of silicon particles made the voids larger and the reflux more, so the corresponding impurity content at the bottom was higher, which made the length of the red zone at the bottom of the mixed phase higher. The average minority carrier lifetime of fused quartz squares was 6.35 μs after removal of the red zone, while that of mixed seeds squares was 6.92 μs. The difference between them was 0.57 μs. This further shows that the minority carrier lifetime and overall quality of the mixed seeds seeded ingot were significantly better than those of the fused quartz particles seeded ingot.

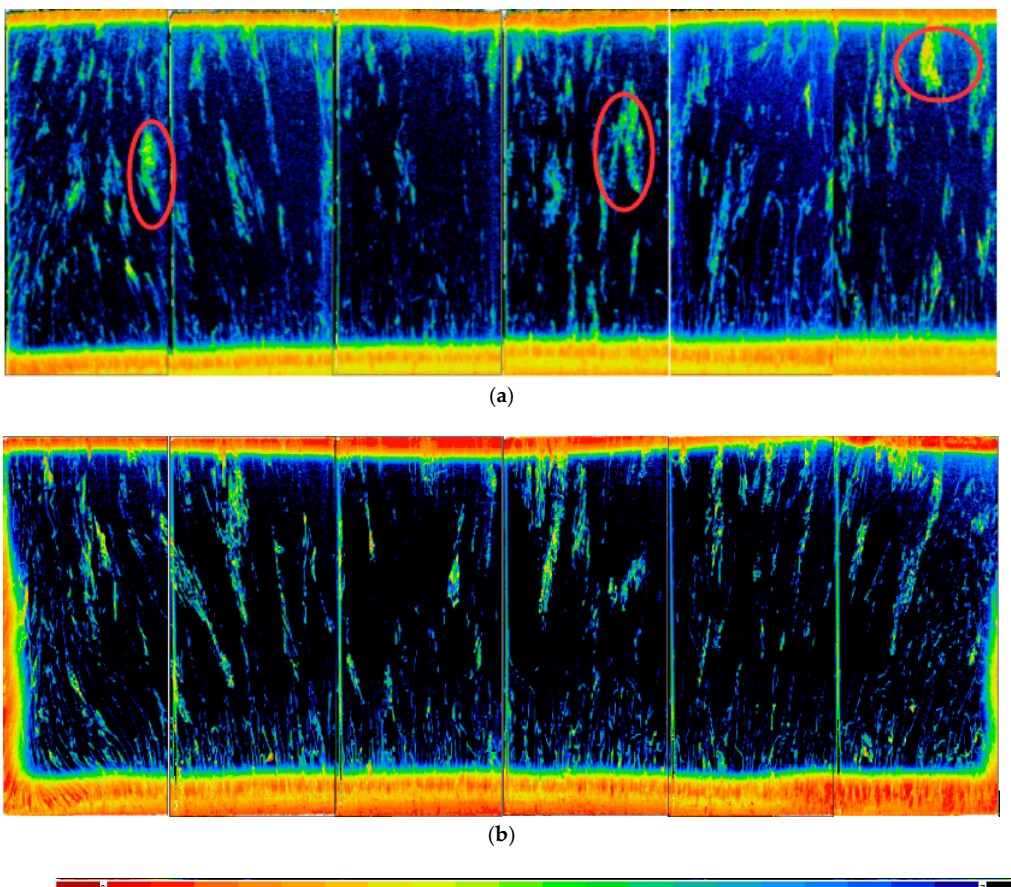

**Figure 7.** Longitudinal minority carrier lifetime mapping of the bricks B13–B18. (**a**) fused quartz particles seeded ingot and (**b**) mixed seeds seeded ingot. (red colour indicates the low lifetime regions and blue indicates the high lifetime regions).

### 3.4. Effect of the Mixed Seeds on the Dislocations

Due to thermal stress and different orientations, dislocations occur during the crystallization process, and the initial dislocations are the sources of other dislocations. During the cooling process after solidification, dislocation density also increases rapidly. Dislocations induce deep concentration centers in the conduction and valence bands of silicon to become composite centers of electrons and holes, which seriously affect the electrical and photoelectric properties of multicrystalline silicon solar cells. Therefore, we need to reduce the formation of dislocations during crystal growth.

Figure 8 shows PL diagrams of wafers from the C15 square at the bottom, middle and top of the two ingots. It can well characterize the grain morphology and dislocation distribution in the cut wafers. The dislocation density in the wafer is expressed by the percentage of dislocation area to the total area of wafer, Rd. The Rd of the fused quartz-seeded ingot was 3.74%, and that of mixed seeds-seeded ingot was 2.96%. By contrast, the average dislocation density of the mixed seeds-seeded ingot was reduced by about 20.9%.

For the fused quartz seeded ingot, there were some dislocation clusters in the PL diagram (shown in the red circle), while the grain size of the mixed seeds-seeded ingot was smaller and more uniform, and the dislocation clusters were almost invisible. Although there were contact nucleation and gap nucleation in both nucleation processes, the latter had larger and more uniform gap nucleation space due to the melting of silicon particles, which had a much smaller impact on the overall nucleation. So, there are two more reasonable nucleation modes and their proportion. On the one hand, the generated gap can better release thermal stress. On the other hand, more fine and uniform initial grains were

formed, which provided a large number of grain boundaries to inhibit the proliferation and diffusion of dislocations [15], thus reducing the dislocation density.

With the growth of the ingots, dislocations (black region) in the ingots proliferated and diffused along the direction of growth. In the mid-growth stage, the dislocation density in the ingot seeded by mixed seeds increased (shown in Figure 8e), but the uniformity of grain size remained at a high level. However, the dislocation density of the fused quartz-seeded ingot in Figure 8b was significantly higher, and most dislocations were concentrated in areas with fine grains. This is because the growth rates of grains with different sizes were different; large grains are continuously extruded and eroded by surface energy, resulting in additional extrusion stress, which lead to more dislocations [15].

At the later stage of crystallization (Figure 8c,f), the size gap between grains gradually enlarged, and the growth competition among grains became more and more fierce, resulting in more extrusion stress. The cumulative release of stress resulted in the rapid growth of dislocations in fused quartz-seeded ingots.

The PL diagram and $R_d$ values show that mixed seeds crystallization can better inhibit the generation and multiplication of dislocations and dislocation clusters. The results also correspond well to the minority carrier lifetime distribution in Figure 7.

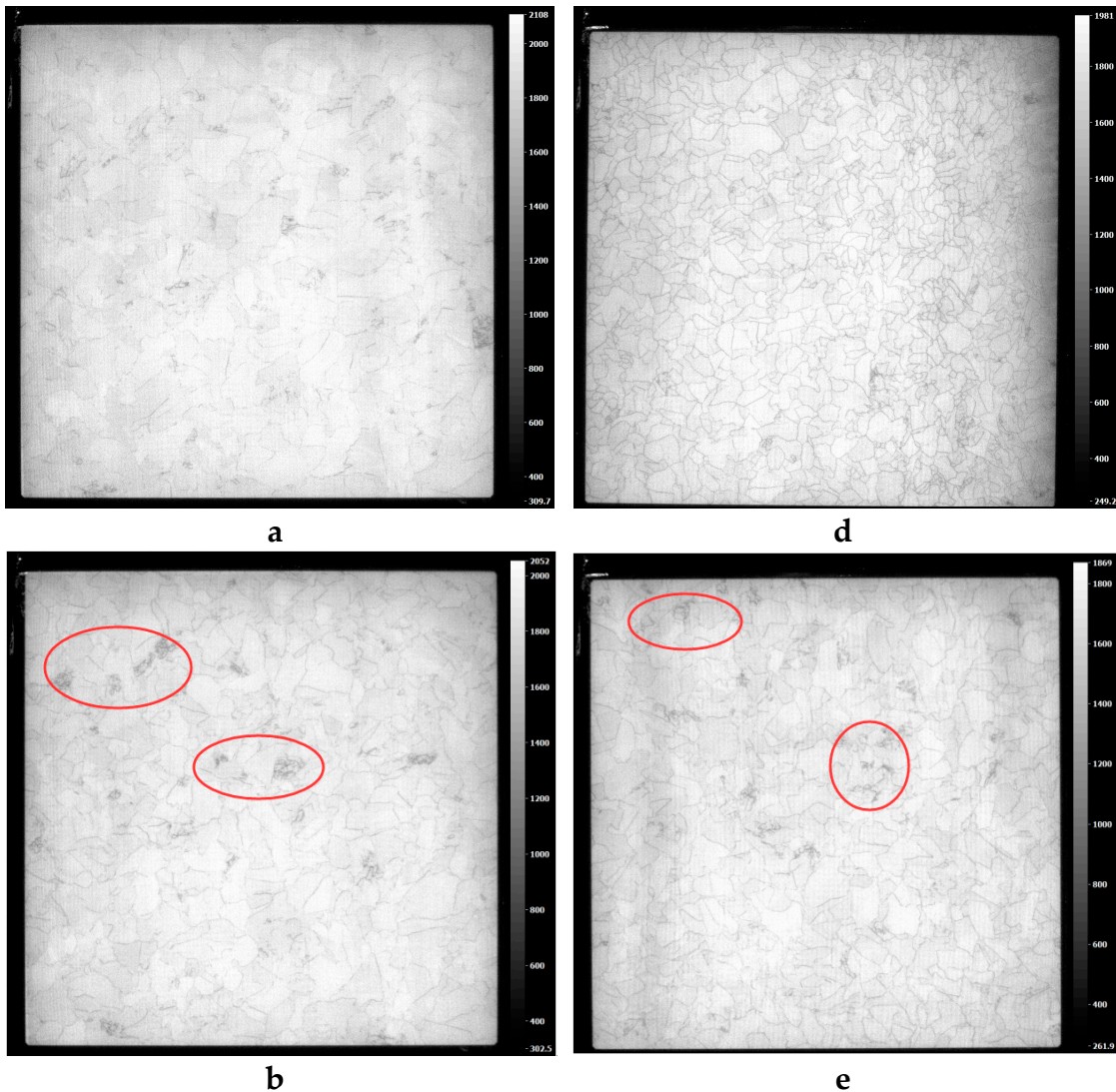

**Figure 8.** *Cont.*

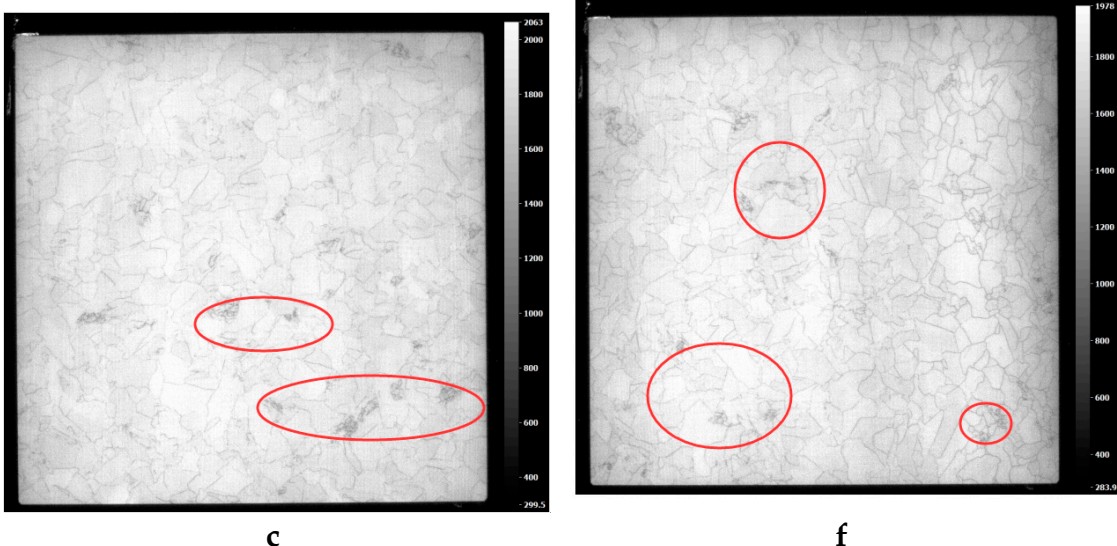

**Figure 8.** PL mappings of the wafers from bottom, middle and top parts of brick C15 from the ingots grown with (**a**–**c**) fused quartz seeds and (**d**–**f**) mixed seeds.

### 3.5. Effect of the Mixed Seeds on the Cell Efficiency

Figure 9 shows the efficiency distribution of solar cells. The average efficiency of solar cells made of silicon wafers with different seeds are 18.60% and 18.69%, respectively. Compared with fused quartz particles, the solar cells made from mixed seeds-seeded ingots not only have higher average efficiency, but also have narrow distribution. The preliminary experimental results showed that using the mixed seeds as the seed layer could improve the cell efficiency of high performance multicrystalline silicon under the same casting conditions. Cell efficiency can also reflect the quality of silicon ingots, such as minority carrier lifetime, dislocations and impurities. This result is in good agreement with the minority carrier lifetime distribution shown in Figure 7 and the dislocation distribution shown in Figure 8.

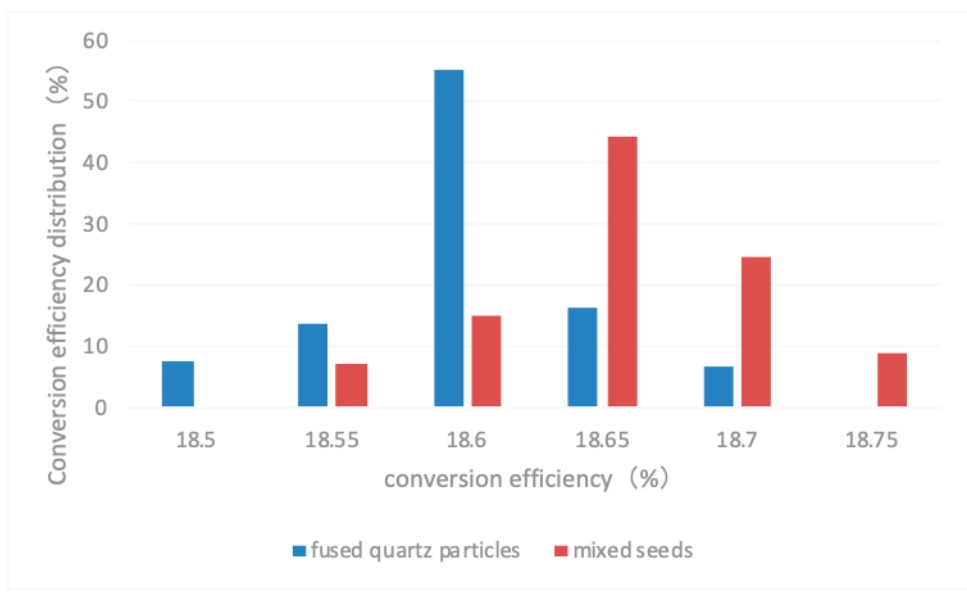

**Figure 9.** Conversion efficiency distribution of the solar cells.

## 4. Conclusions

Multicrystalline silicon ingots were grown using fused quartz particles/silicon particles as mixed seeds. We found that:

(1) Mixed seeds exhibit a great crystallization effect. The average initial grain size was reduced by 17.0% compared with fused quartz crystals, and the uniformity of grain size was improved. Ingot growth using mixed seeds could effectively reduce the dislocation density and improve the quality of ingots.

(2) The average dislocation density of mixed seeds ingots was about 20.9% lower than that of fused quartz seeded ingots. In addition, the minority carrier lifetime was higher, the distribution was more uniform and the average cell conversion efficiency was increased by 0.1%.

We conclude that using fused quartz particles and silicon particles as mixed seeds to assist the growth technology of multicrystalline silicon is beneficial to the preparation of high-efficiency multicrystalline silicon solar cells.

**Author Contributions:** "conceptualization, Y.C. and X.H.; methodology, Y.C.; software, Y.C.; validation, Y.C. and C.M.; formal analysis, Y.C.; investigation, Y.C. and C.M.; resources, Y.C. and C.M.; data curation, Y.C.; writing—original draft preparation, Y.C.; writing—review and editing, X.H.; visualization, Y.C.; supervision, X.H.; project administration, X.H.; funding acquisition, X.H.".

**Acknowledgments:** This work was partly supported by the Program for Changjiang Scholars and Innovative Research Team in University (PCSIRT) [Grant number IRT1146]; a project funded by the Priority Academic Program Development of Jiangsu Higher Education Institutions (PAPD); a project funded by the Natural Science Foundation of Jiangsu Higher Education Institutions [Grant number 13KJB430016]; and a project funded by the Natural Science Foundation of the Jiangsu Province [Grant number BK20141460].

**Conflicts of Interest:** The authors declare no conflict of interest.

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
