# Peer review of "The Artificial Mixed Fused Quartz Particles and Silicon Particles-Assisted High-Performance Multicrystalline Silicon"

_crystals, doi:10.3390/cryst9060286_

Round 1

Reviewer 1 Report

Dear authors,

you have investigated the performance of the mixed seed variant in comparison to the single fused quartz seeding and this field is of high interest in the PV community. However, in my opinion the fused quartz reference doesn't reflect the actual technical standard achievable by this method. All descriptions like grain structure image, dislocation areas and so on looks better than a conventional mc silicon without seeding, but worse than other published results with quartz seeding(.eg. in

Kupka, I.; Lehmann, T.; Trempa, M.; Kranert, C.; Reimann, C.; Friedrich, J., J. Crys. Growth 465 (2017), p. 18–26. or Zhang, H. et al. 2016, J. Crys. Growth, Vol. 435, pp. 91–97.)

So the benefit of the mixed variant to the single variant is not clear for me.

Maybe you can generate more data, like the random grain boundary fraction to compare your both experiments with other ones shown in litertaure.

Some further general remarks:

- the idea of mixing SiO2 with Si particles is not new. You should refer to Wong, Y.T. et al. 2014, J. Crys. Growth,Vol. 404, pp. 59–64.

- fig 1: a) looks like a photograph of the complete crucible, while b) looks like an mikroscope image of only the crucible bottom. Please use the same image type and same image area and insert a scale bar

-fig.3: a vertical grain structure image has not the highest value. nevertheless, the differences in grain structure are not clearly visible in these images. A scanning-image (e.g. of other bricks) should be better in contrast. Scale bar!

-fig. 4:  scale bar !

-lines 122-133: the theory of back-diffusion should be valid for both seeding types. Hence the Si seeds in the mixed layer are melted, there also only quartz particles remaining. Please explain more in detail where then the difference in redzone comes from.

- please take care about a uniform style of your references. In many cases the year is missing.

Author Response

Thank you very much for your review comments!

I have uploaded it as a PDF file.

Reviewer 2 Report

The manuscript with entitle “The Artificial Mixed Fused Quartz Particles and Silicon Particles-Assisted High-Performance Multicrystalline Silicon”, demonstrates the way to improve the quality of hpmc-Si ingot using full melting process using artificial seeding technique, especially, the authors proposed to control a grain size by controlling the morphology of the coating, which closely relates to the grain boundary type of hpmc-Si. The paper is understandable in general, but the journal-style scientific writing providing a clear, logical, and connected story-line is mandatory. In addition, pls make sure a paper is in the best form possible. All images should be modified as well as its caption.

There are many points that have to be addressed and/or clarified before it can be considered for publication. Especially Figures 5, 6, and 7, which contain important information, has to be highlighted and emphasized in a better way. English scientific writing is required.

   In order to be accepted for publication, the following major aspects should be thoroughly revised,

Abstract: Pls revise the abstract to be more informative and pls show the originality of this paper (main mechanism of why mixed particles reduce dislocations and thereby an improved quality of ingot.

Introduction: Pls add more references regarding the advantages of fused quartz particles for the grow of Si ingot applications to the reader

            What is the motivation of this study? My understanding is that; fused quartz particles with full melting process provided high yield but a low quality of ingot so that this study demonstrated how to enhance the quality of mc-Si with comparable yield? Or higher yield? Pls explain..

            Line 50: “the quality of crystal was improved.”…….what is a measure? Lifetime? PL intensity? Or in terms of conversion efficiency of solar cells?

            Line 48: “ Full melting process was used and then we found that the grown crystals had higher quality”…..crystals had higher quality than  ingot using fused quartz particles alone? Or ingot using si particles ?. Pls provide the comparison

 Experimental, 2.1: In this paper, the experimental part is highly important to show the reader how to produce high quality Si ingot using mixing of fused quartz particles and Si particles. Pls explain in more details. For examples;

            -authors may sketch the procedure in 2D or 3D to present the steps of the experiment, seed placing-Si3N4 coating on seeds-coating on the wall, and so on.

            -Pls write this in scientific way…… “a size of 1040 59 *1040 *540mm3”…

            - before placing mixed seeds, Did authors apply the SiN coating layer?

            -Figure 1a: pls explain more about this image

            - Cooling step is essential for crystal growth, pls provide more information of the cooling rate (fast or slow) in this section.

Experimental, 2.2: Line72-74: “The minority carrier lifetime of six squares B13-B18 was scanned by the minority carrier lifetime instrument (Semilab, WT-2000)”….authors mean the Si block was scanned by WT-2000? Or the cut Si wafer was measured by WT-2000?

            - Line71:  6×6 squares mean………6 inch? Pls mention the units or correct scale.

            - Line75-76:  pls explain more about the PL measurement, at Si band or dislocation-related (defect) band ?

            - Line76-77: “Fabrication and performance testing of the solar cells were performed by Yangzhou JA Solar Technology Co., Ltd.”……..authors mean the solar cell device was fabricated using these obtained wafers ? If so, pls explain more about the type of solar cells, aluminum full area back surface field solar cells?

Results, 3.1: Line 85-88: “In this experiment, there were also two kinds of nucleation modes. When the silicon particles …….… fused quartz particles”.

Please show the evidence of the nucleation modes, by showing some pictures or it could be some dislocations in the PL images of the Si wafers at the bottom near seeds. What is the result of the interspace? It led to dislocations? Pls make it clear.

Figure 3: Pls add explanation both in the caption and at the images; i.e. the lighter color means scratches? or dislocations? and throughout

Results, 3.2: Line 91: “the grains on the left side are relatively loose and nonuniform”…the left side means Fig. 3b or the left side of each image? Pls make it clear. Authors may put some arrows to indicate where the position of specific grain boundary is.

Figures 4 and 6: Pls provide high-quality images and provide scale And throughout

Figure 5: Pls provide colour scale; red= low lifetime, blue=high lifetime, and pls explain the differences between Figs. 5a and 5b. and throughout

Figure 6: Pls explain what the red circles are? Would it be better the compare the PL distribution of the wafers obtained from the same ingot but different positions?

Results, 3.3: Pls discuss more in this section, i.e. authors may put some indicators in the Fig. 5, where the waterfall is. Why red-zone of the ingot in Fig. 5b is higher, and etc

            -Authors mentioned that the lifetimes after removal of red zone were 6.35 µs and 6.92 µs for fused-quartz-made and mixed-particles-made ingots, respectively. To the reader view, it was no difference between them. It is better to explain in terms of the difference in the lifetime distributions related to the dislocation present.

            -Since the overall lifetimes of both ingot are the same, my suggestion is; would it be better to compare the microstructure (lifetime map) of sample from different positions (near crucible, middle of ingot, top and bottom)

Results, 3.5: Pls consider to revise this part, and pls modify the graph, probably just choosing the cells from cell positions for both ingot and compare the conversion efficiency.

References: Pls check the format. In the PDF form, there are several mismatch forms of the reference number and the text.

Author Response

(The authors gave the same response as above.)

Reviewer 3 Report

Line 60: The unit "mesh" for particle size is not generally used, therefore it might be good to add the size in micrometer scale. From a scientific point it would be better to give information about the size distribution, not only one value.

The photograph in Fig. 1 (a) does not give any information since no structures can be identified. I assume that one bottom corner of the crucible is shown, but I cannot see anything related to the morphology. Please choose another picture. Furthermore, please add a scale to the pictures.

The title of paragraph 2.2. is "casting process and analysis", but the whole information about the process is just one small sentence in line 70. Since the core of the paper is the seeding procedure which is highly depending on thermal flux, temperature fields and growth properties, a description of the process or at least a summary of the important parameters influencing the seeding should be given such that the reader can understand the general conditions.

In a similar way, a rough model of the nucleation process as mentioned in paragraph 3.1 would give a more scientific insight into the chosen process. A sketch of the described idea of melt in interspaces would give a better understanding of the author's arguments.

You refer in paragraph 3.2 (line 96) to a statistics of grain size distribution, but no data or diagram is shown. You just state the "average grain size". It would be good to either show a diagram for the grain size distributions, or a description of the way you calculate the average grain size.

The quality of the pictures in Fig. 3 is rather poor. Please check if surface pictures with better contrast are available. Additionally, no scale or information is given for the width and length of the scans.

Fig. 4 - analog comment as to Fig. 3

Paragraph 3.3., line 120: What exactly do you mean with "the color of red zone at the bottom of silicon ingot cast with mixed seeds was darker,..."? To which part of the zone of reduced minority carrier lifetime at the bottom part do you refer? The referenced paper of Gao referes to the indiffusion of iron from the melt into the remaining seed. But in the described method, there is no seed preserved from melting execpt the quartz particles. For the region with quartz particles, a minority carrier lifetime cannot be measured. Please clarify how the model of Gao explains the observed lifetime pattern in this case.

Fig. 5 - again please add a scale for the lifetime and information about the length and height

In lines 150 - 151, you refer to a contact nucleation as a first and gap nucleation as a second type of nucleation as referred to in paragraph 3.1  You seem to correlate the two types with the appearance of dislocations. But by just looking at the shown pictures in Fig.6, it is not possible for the reader to understand this correlation since no other data is provided. Please add relevant data such that the correspondence of the type of nucleation and the wafer quality becomes clear.

Fig. 6: It is very hard to see the relevant structures in a print out. Please try to enhance the contrast in the PL images.

In line 174 you refer to "chips". It is unclear what you mean with this.

In Fig. 7. there are two different distributions of cell efficiencies shown. Since you state the the distribution for mixed seed is narrower, please add more bins for higher efficiencies, but at least a bin for 18.75 %

It would be interesting for the discussion to understand the influence of the SiN coating on the seeding procedure. Have you varied the SiN particle size in order to create similar "gap nucleation conditions"? An assessment of this condition could be helpful.

Author Response

(The authors gave the same response as above.)

Round 2

Reviewer 1 Report

Dear authors,

I am not really satisfied by all your comments/corrections.

I am still not convinced that your mixed particle method has a significant benefit.

The shown values for lifetime, dislocation clusters and efficiency are quite close to each other for both variants and should (at least partly) within the margin of error of the measurements methods.

Further, I ask me if you could achieve the same seeding effect if you use less SiO2 particles in order to produce gaps between them...

Nevertheless, you see a (small) difference in your results and therefore it is ok for me.

Concerning the back-diffusion model, your explanation is still not clear. You mention that the impurity reflux from melt towards the bottom should be larger if there are more voids in the seed layer. But where the impurities diffuse to and stay? What is the driving force for this diffusion? In this case you don't have any pure silicon (like the seed plate in Gaos quasi mono case) and it is unlikely that impurities diffuse into the quartz particles due to the extremely low diffusion coefficient of metals in quartz.

I think it is more realistic that there is a stronger diffusion from the crucible bottom through the porous mixed seed layer into the ingot in comparion to the more dense single quartz layer.

Basic things like adding references of people who has the same idea years ago as well as providing high quality images including scale bars should be naturally during publishing a paper.

Best regards,

Reviewer

Author Response

(The authors gave the same response as above.)

Reviewer 2 Report

Page 1, line 22: Monocrystalline silicon…>>> monocrystalline silicon

Page 2, line 60: :silicon particles=1:1)…..>>> :silicon particles = 1:1) pls provide space between =

Page 2, line 61-62: a size of 1040*1040 *540mm3…..>> 1040 mm × 1040 mm × 540 mm

All pages: Pls use the same font and same color

Page 1, Fig. 1: Abour the caption, pls confirm the format with the journal. Could it be….Morphology of the crucible with (a) fused quartz seeds, and (b) the mixed seeds. ???? Pls check!

Page 3, Lines 74-76: 10k stands for ??? Authors mean Kelvin? Or °C? pls check the journal unit format!  “In order to avoid dendrite nucleation at the bottom, the undercooling at the bottom is controlled

below 10k, so the cooling rate at the bottom of crucible is adjusted to about 3k/min, and the growth rate of  silicon ingot is 12mm/h.” Pls modify this sentence. It is not necessary to provide too many paragraphs, authors may wrap them up to only paragraph for subsection 2.2, and font colour should be the same.

All captions for figure: pls check the font size and format!

Page 3, line 3: In Ding et al.[11]'s paper,……>>> In Ding et al., 's paper[11],……et al should be italic

Page 3, Fig. 3: the explanation in Fig. 3 is too small, and difficult to read. Pls use different font colour and possibly bigger size, including pls explain in the caption again.

Page 4, Fig. 4: Pls explain figure 4 in details. The crucible pictures were too small, whereas the particle legend was too large. Pls modify this part.

Page 4, Subsection 3.2 and Fig. 5: pls explain if these photos were taken from the top view or side view of the ingot? The seen grain boundaries and dislocations (white colour in Fig. 5b) are along the growth direction or perpendicular to seeds? Or pls explain more.

Page 4-5, Figs. 5 and 6: (a)fused quartz particles seeded ingot(b)mixed seeds seeded ingot.…This should not be in the figure, it should be in the caption. What is the white colour in Fig. 5b?

Page 6, Lines 131-146: Pls check the font.

Fig. 7: pls explain in the caption that red colour indicates the low lifetime regions and blue indicates the high lifetime regions.

Page 6, Line139: (as shown in the figure)…..>>> as shown in the Fig. 7

Page 7, Line 159: Rd stands for ??? pls explain

Fig. 8: Pls merge all images in Fig. 8 together, then the Fig. 8 will never separate from each other. Pls explain what are in the red circles in Figs. 8c and 8f ? pls write it in both the subsection 3.4 and caption.

References: pls modify the reference by removing the table and [J]. Pls check the journal format.

Author Response

(The authors gave the same response as above.)
